# Phosphorylation of Thr9 Affects the Folding Landscape of the N-Terminal Segment of Human AGT Enhancing Protein Aggregation of Disease-Causing Mutants

**DOI:** 10.3390/molecules27248762

**Published:** 2022-12-10

**Authors:** Jose L. Neira, Athi N. Naganathan, Noel Mesa-Torres, Eduardo Salido, Angel L. Pey

**Affiliations:** 1Instituto de Investigación, Desarrollo e Innovación en Biotecnología Sanitaria de Elche, Universidad Miguel Hernández, 03202 Elche, Spain; 2Instituto de Biocomputación y Física de Sistemas Complejos–Unidad Mixta GBsC-CSIC-BIFI, Universidad de Zaragoza, 50018 Zaragoza, Spain; 3Department of Biotechnology, Bhupat & Jyoti Mehta School of Biosciences, Indian Institute of Technology Madras, Chennai 600036, India; 4Departamento de Química Física, Universidad de Granada, Av. Fuentenueva s/n, 18071 Granada, Spain; 5Center for Rare Diseases (CIBERER), Hospital Universitario de Canarias, Universidad de la Laguna, 38320 Tenerife, Spain; 6Departamento de Química Física, Unidad de Excelencia en Química Aplicada a Biomedicina y Medioambiente e Instituto de Biotecnología, Universidad de Granada, Av. Fuentenueva s/n, 18071 Granada, Spain

**Keywords:** conformational landscape, spectroscopy, circular dichroism, NMR, phosphorylation, mutation, genetic disease, protein misfolding, primary hyperoxaluria type I, statistical mechanics

## Abstract

The mutations G170R and I244T are the most common disease cause in primary hyperoxaluria type I (PH1). These mutations cause the misfolding of the AGT protein in the minor allele AGT-LM that contains the P11L polymorphism, which may affect the folding of the N-terminal segment (NTT-AGT). The NTT-AGT is phosphorylated at T9, although the role of this event in PH1 is unknown. In this work, phosphorylation of T9 was mimicked by introducing the T9E mutation in the NTT-AGT peptide and the full-length protein. The NTT-AGT conformational landscape was studied by circular dichroism, NMR, and statistical mechanical methods. Functional and stability effects on the full-length AGT protein were characterized by spectroscopic methods. The T9E and P11L mutations together reshaped the conformational landscape of the isolated NTT-AGT peptide by stabilizing ordered conformations. In the context of the full-length AGT protein, the T9E mutation had no effect on the overall AGT function or conformation, but enhanced aggregation of the minor allele (LM) protein and synergized with the mutations G170R and I244T. Our findings indicate that phosphorylation of T9 may affect the conformation of the NTT-AGT and synergize with PH1-causing mutations to promote aggregation in a genotype-specific manner. Phosphorylation should be considered a novel regulatory mechanism in PH1 pathogenesis.

## 1. Introduction

The native state ensemble of proteins contains folded, partially folded, and unfolded microstates populated according to their Gibbs free energy, which can be regarded as macroscopic substates [1]. These different microstates (and macroscopic substates) may play different functional roles, affecting ligand binding or activity (i.e., allosterism), the ability to be transported to different organelles (e.g., peroxisomal proteins can be transported as native proteins but mitochondrial proteins may require unfolding), or efficient degradation [2,3,4,5,6,7]. Mutations and post-translational modifications (PTMs, such as phosphorylation or acetylation) may reshape this conformational landscape, affecting multiple features, such as conformational stability, activity, functional cooperativity, intracellular degradation, and protein:protein interactions [8,9,10,11,12,13,14]. In some cases, the redistribution of microstates in the native state ensemble caused by site-specific PTMs can be regarded as disordered-ordered transitions with functional implications [15,16]. Importantly, these events (mutations and PTMs) can be associated with human disease [11,17,18,19,20,21,22,23]. Therefore, it is important to understand the biophysical mechanisms by which mutations and PTMs affect protein conformational ensembles and how these may translate into variable phenotypes between individuals with the same genotype.

Primary hyperoxaluria type I (PH1) is a rare metabolic disease caused by mutations in the *AGXT* gene causing loss-of-function in the alanine:glyoxylate aminotransferase (AGT) enzyme [23,24]. The AGT enzyme is involved in the detoxification of glyoxylate in liver peroxisomes [24]. This disease shows a recessive and autosomal pattern of inheritance, and over 200 mutations in the *AGXT* gene have been reported in PH1 patients [23,24]. Interestingly, PH1-causing mutations are found with higher frequency on the *AGXT* minor allele (LM), which contains (among other changes) the P11L (polymorphic) variation that predisposes towards the deleterious effects of additional mutations and is present in 20% of human alleles and 50% of PH1 patients [23,24,25]. The most common PH1-causing mutations are LM-G170R and LM-I244T (that are found in cis with the LM), accounting for 40% of the alleles found in PH1 patients in European and American registries [26,27]. The LM-G170R primarily causes AGT loss-of-function through mitochondrial mistargeting of the enzyme (where the enzyme is metabolically useless), while LM-I244T leads to protein aggregation and inactivation in the peroxisomes [23,28,29,30]. The final destination of the AGT protein, due to these two mutations (aggregation vs. mitochondrial mistargeting), is in a delicate balance between structural destabilization upon mutation and enhanced interaction with molecular chaperones and other elements of the protein homeostasis network [7,29,31,32,33]. Interestingly, mutations at the G170 and I244 sites can exchange their main mechanisms (mistargeting vs. aggregation), depending on the local destabilizing effects of mutations [19].

Structurally, the AGT protein forms native dimers with a two-domain structure [34] (Figure 1A). The N-terminal domain (residues 1–282) contains a short N-terminal sequence (NTT, residues 1–21; we will refer to this peptide in isolation as NTT-AGT peptide) in a solvent-exposed and extended conformation that grabs the adjacent monomer in the dimer (Figure 1A) and is known to be important for the proper folding and stability of the AGT dimer [35]. Although the NTT in the native AGT dimer shows an extended conformation, it has been proposed that the P11L polymorphism affects its conformation by increasing its propensity towards helical conformations competent for mitochondrial mistargeting of the enzyme [36]. The rest of the N-terminal domain contains the active site of the enzyme and most of the dimerization interface [34]. The C-terminal domain contains the peroxisomal targeting sequence (PTS), which is one of the weakest signals found in the human peroxisomal proteins that interact with the Pex5 peroxisomal receptor [7,37].

A phosphorylation event at T9 of human AGT has been reported in a high-throughput analysis of the human liver phosphoproteome [38]. This site is located in the NTT of human AGT (but not in most mammalian AGTs), in the vicinity of the highly conserved P11 (Figure 1). We hypothesize that phosphorylation of T9 may affect the conformation of the NTT, and consequently, the aggregation or mistargeting of AGT in PH1. In addition, the benign missense T9N variant (rs115014558) has been reported in human population, with an allele frequency ranging between 0.3 and 1% (https://www.ncbi.nlm.nih.gov/clinvar/variation/204016/, accessed on 1 November 2022). Although T9 is far from those mutations at G170R and I244T (Figure 1B), the long-range propagation of conformational and stability effects may occur in the structural ensemble of AGT upon phosphorylation or mutations [19]. Thus, conformational alterations due to phosphorylation at T9 may have unexpected effects on the physiological regulation of AGT activity, as well as in the frequent mistargeting and aggregation phenotypes in PH1. In addition, phosphorylation of the T9 site of human AGT may represent a very specific regulatory mechanism, since this residue is highly specific of the human protein among close mammalian orthologues (Figure 1B). 

In this work, we investigated the potential effects of phosphorylating T9 using the phosphomimetic mutation T9E, and in some instances, the mutation T9A as a control. We used two complementary approaches. First, we investigated, in great detail, the conformational preferences of the WT NTT-AGT peptide (the first 21 residues of AGT) in the absence/presence of the mutations T9E and P11L. We used spectroscopic methods (circular dichroism (CD) and NMR spectroscopies) in the absence or presence of trifluoroethanol (TFE). These experimental analyses are combined with statistical mechanical approaches to investigate the conformational landscape of the NTT-AGT peptides. We complemented these studies with those on the full-length AGT protein in the absence or presence of the LM polymorphism and the pathogenic mutations G170R (LM-G170R) and I244T (LM-I244T) and analyzed their effects on the activity, stability, and aggregation of the AGT. 

## 2. Results

### 2.1. The T9E and P11L Mutations Synergistically Reshape the Conformational Landscape of the NTT-AGT Peptides by Enhancing Structural Order

#### 2.1.1. Conformational Transitions of NTT-AGT Peptides Studied by Optical Spectroscopy 

The NTT-AGT region adopts an extended conformation in the structure of AGT that wraps the adjacent monomer in the dimer (Figure 1A). A previous study showed that the NTT-AGT peptide has a weak helical propensity, which was slightly increased by the addition of TFE (50%, 7 M) [36]. Since the NTT-AGT peptide has four prolines (P10, P11, P17, and 21), is clear that its population of helix structure should be weak and involve those residues distant from the prolines. In such previous studies, the mutation P11L had no clear effect on the conformational propensity of the NTT-AGT peptide. We must note that these authors used similar but not identical conditions to ours. First, the pH was 7.0 (we used pH 7.4), which may lead to different fractions of H4 in a protonated state (note that the p*K_a_* for His residues in about 6, if non-perturbed by the peptide structure). Second, they attached a Cys residue at the C-terminal of the peptide (while we used a Y to determine the peptide concentration accurately, see Section 4.2.1) that, under non-reducing conditions, may lead to the formation of an intermolecular S-S bond. 

To determine whether the mutation T9E could affect the propensity of the NTT-AGT peptide to adopt structured conformations, we characterized the peptides containing either T9E and/or P11L by far-UV CD (and NMR, Section 2.1.2) spectroscopy. In the solution, all peptides showed little or no evidence for ordered conformations, thus resembling random coil conformation (i.e., with a strong and negative dichroic band centered at ~200 nm; Figure 2A [39]) with a weak tendency in peptides containing T9E to locally adopt ordered conformations (Figure 2A and Figure 3, Appendix A). The addition of a high TFE concentration (3.9 M) led to an increase in the ordered structure in the presence of p.P11L (i.e., a decrease in the dichroic signal at 200 nm and an increase at 225 nm, possibly resembling some α-helical or β-turn-like content [39]), and this effect was particularly large when the polymorphism was combined with the phosphomimetic mutation T9E (Figure 2B,C). We monitored the changes in ellipticity at 225 nm, instead of the more common 222 nm, to follow the increase of helicity, because of the minimum of the shoulder was observed at the highest concentration of co-solvent for some of the peptides (Figure 2B for P11L and the double mutant). Since we are determining Δ*G*-values, from which we can estimate the helical population (and we are not using absolute values of ellipticity at a particular wavelength to determine helicity), the wavelength where we monitored the TFE-transition at is not important, as long as the signal-to-noise ratio is good enough. These effects were less evident by NMR spectroscopy (Section 2.1.2., Figure 3 and Appendix A), suggesting a transient nature for TFE-induced ordered structure, which was further supported by the non-cooperative thermal denaturation of the peptides under these conditions (Appendix A). The fact that the ellipticity was increased (in absolute value) in all peptides at 225 nm suggests that the enhanced folded population could resemble an α-helix-like conformation, although the absence of a significant increase in the ellipticity at 208 nm did not allow us to exclude the enhancement of β-turn-like conformations [39]. At this stage, it is important to pinpoint that the use of TFE in peptides does not induce any non-native-like α-helix or β-turn-like conformations, but rather enhances the populations of those intrinsic structures in the corresponding peptides [40,41]. 

To assess whether these amino acid changes would affect the thermodynamics of the disorder-to-order transition, we performed titrations of the peptides with different concentrations of TFE that supported a two-state-like behavior for these transitions (Figure 2D,E and Appendix A). Global fitting of the CD signals at 200 and 225 nm to Equations (2) and (3) [40,41] (Figure 2D,E) provided similar folding free energies for all peptides (Figure 2G, Appendix A), although the transition for the peptide containing both mutations (T9E and P11L) was more cooperative (i.e., higher m-value), in agreement with a higher degree of ordered structure in the presence of TFE (Figure 2F). Intriguingly, the isodichroic point found in these titrations was ~212 nm (Appendix A), a value somewhat higher than those reported for the random coil-to-α-helix transitions in short peptides (~203 nm) [40].

#### 2.1.2. Conformational of NTT-AGT Peptides Studied by NMR Spectroscopy

To investigate, in further detail, the conformational features of the AGT-NTT peptides, we carried out NMR experiments in aqueous and TFE solutions. We first tested the self-association properties of the peptides in aqueous solution by using translational diffusion measurements. All the peptides (Table 1) in aqueous solution were monomeric, with estimated *R*_h_s close to those obtained for random-coil peptides [42]. The disordered character of all peptides in phosphate buffer was further confirmed by the 1D-^1^H-NMR spectra, which showed a clustering of signals of all the amide protons between 8.0 and 8.6 ppm, whereas the methyl protons were observed between 0.8 and 1.0 ppm (Appendix A). For those kinds of protons, all such values are typical of disordered polypeptide chains [43].

In aqueous buffer, we could not fully assign all the residues in each peptide unambiguously, due to the overlapping and broadening of the amide signals for some residues from the homonuclear 2D ^1^H-NMR spectra. The peptides were mainly disordered in solution, as suggested by two lines of evidence, further confirming the results from the 1D-^1^H-NMR spectra (Appendix A). First, the sequence-corrected conformational shifts (Δδ) of the H_α_ protons [43,45,46] for unambiguously assigned residues were within the commonly accepted range for random coil peptides (│Δδ│ ≤ 0.1 ppm) (Appendix A). Second, no long- or medium-range NOEs were detected, but only sequential ones (i.e., αN (i, i + 1) and βN (i, i + 1)) in the polypeptide patches were fully assigned (Figure 3).

Based on our far-UV CD results (Figure 2A,B), we then carried out the acquisition and assignment of 2D-^1^H-NMR spectra in the presence of 30% TFE (4.2 M). For all NTT-AGT peptides, except for NTT-AGT T9E, the signals of the NH protons became sharper and more spread than in aqueous solution (Appendix A), paving the way to a complete assignment of most of the residues in each peptide. In addition, the presence of 30% TFE resulted in: (a) large negative conformational shifts (i.e. │Δδ│≥ 0.1 ppm), especially around the polypeptide regions P10-L12 and L14-K16, indicating the presence of β-turn- or α-helix-like conformations (Appendix A); and (b) the presence in all the peptides of sequential NN (i, i + 1) contacts through their sequences, further confirming the presence of β-turn-like conformations or nascent helices. However, except in the case of NTT-AGT WT, where medium-range NOEs (αN (i, i + 3)) were observed around Leu14 (Figure 3), no other medium-range NOEs were observed. On the other hand, the spectrum of NTT-AGT T9E in the presence of 30% TFE was very broad (Appendix A), indicating the presence of conformational exchange and hampering assignment (Appendix A). 

Therefore, we can conclude from this section that the NTT-AGT peptide has an intrinsic, natural (but weak) tendency to populate α-helix or β-turn-like conformations, and that this tendency is increased when the mutations T9E and P11L occur concomitantly, as suggested by far-UV CD.

#### 2.1.3. Conformational Landscape of NTT-AGT Peptides Studied by Statistical Mechanical Methods

We then applied the Wako-Saitô-Muñoz-Eaton (WSME) statistical mechanical model to evaluate the effect of the T9E and P11L substitutions on the conformational preferences of the NTT-AGT peptides (Methods and Figure 4). 

We first evaluated the probability of folded residues existing along the sequence of NTT-AGT peptides (Figure 4A). The mutation T9E increased the folding probability along the entire peptide, whereas P11L only slightly increased the probability locally (i.e., at P11), with a synergic effect of both mutations on such folding probability. Secondly, we calculated the free energy levels for the four NTT-AGT peptides along a simple reaction coordinate (i.e., the number of folded residues) (Figure 4B). These calculations suggested that the most stable macrostate for the NTT-AGT WT peptide is the unfolded state (U), and two different partially folded (PF) and a folded (F) state were populated along the folding of the peptide with increasing free energies (particularly the F state). These results also suggest that the PF and F states had different free energy levels, with free energies differing by ~1.5 and ~3 kcal·mol^−1^ (Figure 4B), values close to the overall free energy obtained experimentally for the disorder-to-order transition in the presence of TFE (~2–3 kcal·mol^−1^; Appendix A). This scenario was not quantitatively or qualitatively affected by the P11L mutation alone. The peptide containing the mutation T9E also showed these four macrostates, but strongly stabilized the two most unstable and more folded states found in the NTT-AGT WT peptide, whereas the two mutations together slightly enhanced these effects. A two-dimensional representation, in which the structural order of the N-terminal and C-terminal halves of the NTT-AGT peptide were analyzed, also showed some interesting differences. In the NTT-AGT WT peptide, the two PF states showed quite localized ordered ensembles, with uncoupled folding of the N-terminal and C-terminal sections (Figure 4C, WT). The presence of the T9E mutation had two important effects (Figure 4C, T9E): (1) the structure of the PF macrostates shifted towards more ordered conformations (with the unfolded half now partially folded); and (2) highly ordered substates (close to the F macrostate) were strongly stabilized. 

Overall, these calculations support that both mutations synergize in the stabilization of partially folded states in the NTT-AGT peptide, in agreement with the experimental results described in Section 2.1.1 and Section 2.1.2.

### 2.2. The T9E Mutation Did Not Affect the Overall Conformation and Catalytic Performance of the AGT Protein

As we have shown in the study of the isolated peptides, the phosphorylation of T9 seemed to change the conformational space of the region around this residue; therefore, we wondered whether the phosphomimetic mutation T9E could affect the conformation and activity of the AGT protein. We introduced this mutation, as well as the negative control T9A, in four different backgrounds: two non-pathogenic (WT and LM) and two pathogenic (LM-G170R, primarily associated with mitochondrial mistargeting, and LM-I244T, primarily associated with peroxisomal aggregation). The T9A is a negative control, since it does not introduce the phosphomimic glutamic mutation in the proteins. All 12 possible variants were expressed in, and purified from, *E. coli* using standard procedures [31,32].

The effect of the mutations T9E and T9A on the overall conformation of AGT was determined by dynamic light scattering (DLS) and far-UV CD- spectroscopy (Figure 5A,B). DLS analyses showed that these two mutations did not affect the hydrodynamic volume of the holo-AGT dimer (the average ± s.d. for the radii were: control group: 3.94 ± 0.16 nm; T9E group: 3.82 ± 0.06 nm; T9A group: 3.84 ± 0.11 nm) (Figure 5A). These values are consistent with the molecular dimensions of AGT dimer from X-ray crystallography (with a radius of ~ 4.7 nm) [34]. In addition, all variants showed similar far-UV CD spectra to those of the holo−proteins, further supporting that the mutations T9E and T9A did not affect the overall conformation of the AGT protein (in this case, its secondary structure content) (Figure 5B). Activity measurements also supported that these two mutations did not affect the overall transaminase activity of AGT (the average ± s.d. for AGT activity were, in mmol of pyruvate (Pyr)·h^−1^·mg^−1^: control group: 1.43 ± 0.20; T9E group: 1.57 ± 0.11; T9A group: 1.57 ± 0.29) (Figure 5C).

Additional spectroscopic measurements confirmed that the physico-chemical properties of the AGT active site (particularly, the binding sites for PLP/PMP) were not affected by the mutations T9E or T9A (Figure 6A,B). During the first half of the AGT transamination activity, an amino group is transferred from L-Ala to the PLP covalently bound to K209, forming pyruvate (Pyr) and PMP. This reaction can be followed by particular changes in the absorption and CD spectra of PLP and PMP [47]. All the holo −proteins showed the characteristic near-UV/visible spectra of the WT protein (Figure 6A) with no effects of the T9E or T9A mutations on the maximum wavelength of the visible absorption band (the average ± s.d. for λ_max_ were for all groups 421 ± 1 nm) or the intensity (the average ± s.d. for ε_@λmax_ were, in mM^−1^·cm^−1^: control group: 5.56 ± 0.64; T9E group: 6.13 ± 0.40; T9A group: 5.52 ± 0.72). The PMP formed upon the first half-transamination showed spectral properties in all variants similar to those observed in the WT protein (Figure 6A): the maximum of the near-UV absorption band (the average ± s.d. for λ_max_ were: control group: 331.0 ± 0.7 nm; T9E group: 331.7 ± 0.8 nm; T9A group: 331.5 ± 0.5 nm) or the intensity (the average ± s.d. for ε_@λmax_ were, in mM^−1^·cm^−1^: control group: 7.61 ± 0.60; T9E group: 8.70 ± 1.07; T9A group: 7.83 ± 1.65). Similar experiments carried out by using near-UV CD spectroscopy confirmed that the local microenvironment of the PLP/PMP was not affected by the T9E or T9A mutations (Figure 6B). As holo −proteins, all proteins showed similar near-UV CD spectra: the maximum of the visible absorption band (the average ± s.d. for λ_max_ were: control group: 427.0 ± 2.3 nm; T9E group: 426.2 ± 2.8 nm; T9A group: 425.5 ± 1.5 nm) or the intensity (the average ± s.d. for [*Θ*]_@λmax_ were, in deg·cm^2^·dmol^−1^: control group: 160.5 ± 15.0; T9E group: 163.9 ± 16.8; T9A group: 171.6 ± 13.9). When alanine was added, all proteins showed similar near-UV CD spectra: the maximum of the near-UV dichroic band (the average ± s.d. for λ_max_ were: control group: 311.5 ± 3.7 nm; T9E group: 310.8 ± 1.8 nm; T9A group: 310.0 ± 1.3 nm) or the intensity (the average ± s.d. for [*Θ*]_@λmax_ were, in deg·cm^2^·dmol^−1^: control group: 49.2 ± 3.3; T9E group: 47.7 ± 9.6; T9A group: 50.3 ± 9.6).

### 2.3. The T9E Mutation Did Not Affect PLP or PMP Binding Affinity

To check whether the phosphomimetic mutation T9E could affect the binding affinity for PLP or PMP, we carried out two types of experiments. The affinity for PMP is known to be higher than that for PLP (about 3-fold), and its binding to the apo −protein is extremely slow (it takes about 10 h to reach 95% of saturation, based on [47]). This high affinity allows to the PMP-AGT protein to retain most of the PMP bound during the catalytic cycle [47]. Thus, to compare the apparent affinity for PMP between AGT variants, we used the PMP-AGT form upon the first half-transamination and measured the fraction of PMP released upon filtration through a 30 kDa cut-off filter (Figure 5C). It must be noted that this procedure is semi-quantitative and allows us to identify only those AGT variants with a largely affected affinity for PMP [31,47]. The amount of PMP released is, thus, determined from filtrating a protein solution of holo −proteins that react with a large excess of L-Ala and quantifying the absorbance in the filtrate at 330 nm [31]. The results obtained did not support large changes in PMP binding affinity by the T9E and T9A mutations (the average ± s.d. for % of PMP estimated to be released were: control group: 15.8 ± 2.9; T9E group: 12.6 ± 6.1; T9A group: 13.5 ± 4.7) (Figure 6C).

We then carried our PLP binding experiments by analyzing its binding kinetics under pseudo first-order conditions at 25 °C (Appendix A). This procedure was selected over equilibrium binding titrations, due to the aggregation of some apo-variants during preliminary tests for such equilibrium measurements. As previously noted, this procedure provides reliable data for an association rate constant (*k*_on_), although the value of the *k*_off_ is not reliably determined, since it is close to zero. The experimentally determined *k*_on_ values (obtained from the slope of observed binding rate constants on [PLP]) were determined more accurately and clearly showed very little effect of the T9E and T9A mutations (the average ± s.d. for the *k*_on_ were: control group: 74.9 ± 11.6 M·s^−1^; T9E group: 72.4 ± 11.1 M·s^−1^; T9A group: 75.2 ± 8.3 M·s^−1^) (Figure 7B and Appendix A). The results obtained did not support large changes in PLP binding affinity, due to the T9E and T9A mutations (the average ± s.d. for the *K*_d_ were: control group: 3.9 ± 1.3 µM; T9E group: 5.4 ± 3.5 µM; T9A group: 4.5 ± 2.2 µM) (Figure 7A). As expected, these *K*_d_ values differed from those determined by equilibrium titrations in one order of magnitude [31,47], likely due to the error inherent to the determination of *k*_off_ values. As expected from the limitations of these kinetic binding studies, the results in the determination of the *k*_off_ values showed larger scatter between individual variants (Figure 7C). However, when the results were grouped, we observed that the T9E and T9A mutations had no effect on this parameter (the average ± s.d. for the *k*_off_ were: control group: 3.0 ± 1.5 × 10^−4^ s^−1^; T9E group: 3.4 ± 1.6 × 10^−4^ s^−1^; T9A group: 4.2 ± 0.7 × 10^−4^ s^−1^). 

### 2.4. The T9E Mutation Did Not Affect the Thermal Stability of AGT

Thermal stability of AGT is largely increased upon binding of PLP [32,48] (Figure 8). Interestingly, PH1-causing mutations affect much more the thermal and kinetic stability of the apo-state (note that thermal denaturation is not reversible and no amenable for thermodynamic analysis) [31,32] (Figure 8). Studies of AGT thermal stability showed that the effects of the mutations T9E and T9A were negligible (the average ± s.d. for the Δ*T*_m_ between T9E or T9A mutant and control groups were: apo-T9E group: −0.2 ± 0.8 °C; holo-T9E group: 0.5 ± 0.7 °C; apo-T9A group: 0.3± 1.2 °C; holo-T9E group: 0.3 ± 0.6 °C; note that a negative value would indicate reduced stability upon T9E/T9A mutation). 

### 2.5. The T9E Mutation Enhanced Aggregation of the Disease-Associated Variants in the Apo-State

Enhanced protein aggregation is a common effect of PH1-causing mutations, particularly in the apo-AGT state [29,31,48,49]. Interestingly, since protein aggregation can be decoupled from global conformational stability (i.e., thermal stability) [50,51], we investigated the effect of the mutations T9E and T9A on the aggregation kinetics of holo- and apo-AGT at 37 °C (Figure 9). 

In contrast to experimental analyses described in Section 2.2 and Section 2.3, we observed some effects of the mutations T9E and T9A, which depended on the background variant (WT, LM, LM-G170R, and LM-I244T) (Figure 9A,B). We extracted two different parameters from these experiments: the *extent* (scattering at 400 nm after a 90 min incubation) and *maximal rate* (as the maximal value of the first derivative of scattering vs. time) of aggregation (Figure 9C,D). Both parameters provided similar results. None of the variants showed significant aggregation in the holo-state (Figure 9B–D). Interestingly, the mutations T9E and T9A had no effects on apo-WT, whereas they affected the aggregation of apo-LM, apo-LM-G170R, and apo-LM-I244T in different manners. T9A enhanced the aggregation of apo-LM by ~2.5-fold, while this effect was 10-fold larger for T9E. G170R enhanced the aggregation of AGT-LM by 15-fold, while the addition of T9A had no effect, and the mutation T9E slightly enhanced the aggregation of apo-AGT-LM-G170R. The mutation I244T enhanced the aggregation of apo-LM by 4-fold, whereas the mutations T9A and T9E enhanced the aggregation of apo-LM-I244T by 2- and 3-fold, respectively. 

Therefore, two main conclusions can be drawn. First, the mutation T9E largely enhanced aggregation of apo-AGT in the presence of the P11L polymorphism. At this stage, it is important to remind (Section 2.1.3) that the T9E mutation in the NTT-AGT peptides led to a shift in the conformational landscape, resulting in a higher population of ordered conformations; thus, it is tempting to suggest that the presence of those more ordered conformations could be a previous kinetic step to form aggregated species in the whole intact protein. Second, in the presence of PH1-causing mutations that enhanced protein aggregation, the effect of T9E was milder and depended on the intrinsic aggregation propensity of the AGT background (the lower the propensity of the mutant background, the higher the effect of the T9E mutation).

## 3. Discussion 

Protein phosphorylation is a general mechanism to rapidly regulate different protein functions as a result of changes in physiological conditions [52,53]. However, the role of site-specific protein phosphorylation in the modulation of pathological phenotypes is not well-understood [54,55]. In this work, we present the first report on the consequences of site-specific phosphorylation in human AGT, a metabolic enzyme whose deficit leads to PH1, a monogenic metabolic disease caused by mutations affecting protein stability and trafficking [19,24]. By 14 October 2022, six phosphorylation sites (T9, S81, Y194, Y231, Y260, and Y297) had been reported by high-throughput proteomic approaches in the PhosphoSitePlus database (https://www.phosphosite.org/homeAction.action, accessed on 1 November 2022) [56]. However, no studies on their functional consequences had been reported. We selected the T9 site located in the NTT-AGT, proposed to be involved in the mistargeting and aggregation of AGT in PH1 [35,36]. Our study shows that the conformational preferences of the NTT-AGT are shifted towards ordered conformations by a phosphomimetic mutation (T9E), particularly when it is present in *cis* with the P11L polymorphism commonly found in PH1 patients. However, the NMR findings in aqueous solution (Figure 3) and the shape of the far-UV CD spectra (Figure 2), together with their deconvolution by using the DICHROWEB site [57,58,59]—where the percentages of disordered structure for any the four peptides range between 41 to 74%, and those of α-helix did so between 12 to 17%—indicate that the population of ordered structures was very small in water. Studies on the full-length AGT protein show no effects on catalysis, PLP/PMP binding, or thermal stability. However, the T9E mutation exacerbated the aggregation tendency of different forms of AGT associated with disease in a genotype-dependent manner. We, thus, propose that site-specific phosphorylation might be a factor contributing to genotype–phenotype correlations in PH1, beyond the presence of the two most common disease-associated alleles in the *AGXT* gene.

In an early study, the overall conformation of the NTT-AGT peptide was evaluated at pH 7.0 in the absence or presence of 50% TFE. The results obtained showed a very low tendency of this peptide, even in the presence of the P11L polymorphism, to adopt ordered conformations, based on results from far-UV CD [36]. In our opinion, the most interesting result from this study was the finding of an unnatural mutation (P11L) that enhanced the content in ordered structure of the NTT-AGT peptide, which was also capable of driving a construct fused with a green fluorescent protein (NTT-AGT-GFP) to mitochondria. The authors proposed that the increased content in α-helical structure in the NTT was connected to the mitochondrial mistargeting of the full-length AGT (i.e., in the LM-G170R variant). In our study, we carried out a comprehensive experimental characterization (by far-UV CD and NMR), combined with statistical mechanical calculations, which shows that the phosphomimetic T9E mutation also shifts the conformational equilibrium (particularly in the presence of P11L) from disorder to ordered substates, with some weak content in the α-helix or β-turn like conformations. 

Our results with the full-length AGT protein showed that the T9E mutation did not affect the oligomerization, activity, PLP/PMP or thermal stability of the full-length holo- and apo-proteins. However, it did show effects on the aggregation of apo-proteins, and these effects were specific on the mutant tested. The T9A mutation had milder effects of aggregation. The higher aggregation propensity of apo-LM-I244T and apo-LM-G170R in the presence of T9E actually suggests that this phosphorylation event could modulate the loss-of-function phenotype of these variants inside cells (see, for instance, [19]). The most remarkable effect was found for apo-LM, and this result may have even more general implications for PH1 pathogenesis, since this polymorphic background is 2.5-fold more frequent in patients (about 50% of disease-associated alleles) than in healthy individuals [24]. Therefore, these results suggest that phosphorylation at T9 could be a relevant modulator of genotype–phenotype relationships in PH1 in a large fraction of patients. The milder effects of the T9A are supported by evolutionary analysis, since some mammalian AGT sequences show an Ala residue at this position and may imply some differences in the aggregation propensity of apo-AGT among mammalian species.

Overall, we have characterized the potential effects of phosphorylation at the T9 site of AGT using comprehensive biophysical and biochemical experiments and computational studies. Our results suggest that phosphorylation may modulate the aggregation propensity of disease-associated variants. We will apply similar strategies to understand the role of phosphorylation at different sites of AGT and their potential roles in genotype–phenotype correlations in PH1.

## 4. Materials and Methods

### 4.1. Materials

Ultra-pure dioxane, non-deuterated and deuterated trifluoroethanol (TFE), 2-Amino-2-(hydroxymethyl)-1,3-propanediol (TRIS), 4-(2-Hydroxyethyl)piperazine-1-ethanesulfonic acid (HEPES), L-alanine, pyridoxal 5′-phosphate, glyoxylate, trichloroacetic acid, D_2_O, lactate dehydrogenase (from rabbit muscle), and TSP (3-(trimethylsilyl)-2,2,3,3-tetradeuteropropionic sodium salt) were from Sigma (Madrid, Spain). Ampicillin and isopropyl β-d-1-thiogalactopyranoside (IPTG) were from Canvax Biotech (Valladolid, Spain).

### 4.2. Characterization of the Effects of T9E and T9A on NTT-AGT Peptides 

#### 4.2.1. NTT-AGT Peptides

The first 21 amino acids (NTT) of AGT with a C-terminal Y (amidated), containing T9E and/or P11L mutations (NTT-AGT peptides), were chemically synthesized by GenScript Biotech (Leiden, The Netherlands) with a minimal purity of 95%. The sequence of the peptides were (residues in red are those mutated):WT   MASHKLLVTPPKALLKPLSIPY
P11L   MASHKLLVTP**L**KALLKPLSIPY 
T9E   MASHKLLV**E**PPKALLKPLSIPY
T9E-P11L  MASHKLLV**E**P**L**KALLKPLSIPY

It could be thought that the presence of Y at the C terminus of the peptide could affect the ellipticity at 222 nm, as it has been reported in model peptides [60]; the influence of such Y on the far-UV CD spectrum can be hampered by the introduction of G residues before such an aromatic one. We assumed that the presence of a proline residue preceding the tyrosine could have the same effect as the glycine. Additionally, it must be stated that as we are determining the Δ*G*-values, from which we can estimate the helical population (and we are not using absolute values of ellipticity at a particular wavelength), as long as the changes at a chosen wavelength can be followed with a good signal-to-noise ratio, and the influence of the Y bands at any TFE-concentration for any of the peptides would be the same, and such a presence would not affect the slope of the titration curve.

Peptides were dissolved in ddH_2_O to a final concentration of 0.5–1.5 mM (using a ε_280_ = 1490 M^−1^·cm^−1^ for quantification using UV-visible spectroscopy at 25 °C using an Agilent HP8453 diode-array spectrophotometer (Agilent, Madrid, Spain) and stored at −80 °C. Their solubility and aggregation behavior were routinely checked by means of light scattering measurements.

#### 4.2.2. Far-UV CD Spectroscopy

Far-UV CD spectra were acquired in a a Jasco J-715 (Jasco, Tokyo, Japan) using 1 mm quart cuvettes and thermostatized using a Peltier element. Spectra were registered at 20 °C, in the 190–260 nm range, with 1 nm bandwidth and a time response of 1 s. Four scans at 100 nm·min^−1^ scan rate were acquired and averaged. For experiments using NTT-AGT peptides in the absence or presence of trifluoroethanol (TFE), peptides were prepared at a final concentration of 50 µM in K-phosphate 20 mM pH 7.4 in the presence of 0–8 M TFE. Samples without the peptides in the presence of different TFE concentrations were routinely acquired and subtracted. Samples were incubated at 20 °C for at least 1 h prior to measurements 

To calculate mean residue ellipticities ([*Θ*]_MRE_), we used Equation (1):(1)$${\left[\Theta \right]}_{MRE}=MRW\cdot \Theta_{obs}\frac{MRW\cdot \Theta_{obs}}{10\cdot l\cdot c}$$
where MRW was equal to the molecular weight of the peptides (or similarly for AGT proteins) divided by N-1 (with N = 22 being the number of residues in the peptides and 392 for AGT proteins), *Θ*_obs_ was the ellipticity (in deg), *l* was the path length (in cm), and *c* was the concentration of peptides/AGT in g·mL^−1^. 

To estimate the apparent free energy change associated with TFE-induced conformational transitions of the NTT-AGT peptides, we used a two-state equilibrium model to fit the TFE-concentration ([TFE] in M)-dependent mean residue ellipticity [*Θ*]_MRE_([TFE]) to Equation (2) [40,41]:(2)$${\left[\Theta \right]}_{MRE}\left(\left[TFE\right]\right)={\left[\Theta \right]}_{MRE,TFE}+{\left[\Theta \right]}_{MRE,buffer}\cdot \frac{exp^{\frac{-m\cdot \left(\left[TFE\right]-C_{m}\right)}{R\cdot T}}}{1+exp^{\frac{-m\cdot \left(\left[TFE\right]-C_{m}\right)}{R\cdot T}}}$$
where [*Θ*]_MRE,TFE_ was the mean residue ellipticity of the TFE-induced conformational transition (i.e., at very high TFE concentration), [*Θ*]_MRE,buffer_ was the mean residue ellipticity of the peptides in buffer, m describes the cooperativity of the transition (in kcal·mol^−1^·M^−1^), *C*_m_ was concentration for half-transition (in M), and *R* was the ideal gas constant (1.987 cal·mol^−1^·K^−1^). To provide an accurate estimation of *m*- and *C*_m_ values, we simultaneously fitted the results from 200 and 225 nm using Equation (2). Using the linear extrapolation method, the folding free energy in the absence of TFE (ΔG) can be calculated using Equation (3):ΔG= m· C_m_(3)

Thermal scans of the AGT-NTT peptides were carried out under the same conditions (peptide concentration and buffer) by acquisition of [*Θ*]_MRE_ at 200 or 225 nm from 5 °C to 85 °C or 95 °C at a scan rate of 1.5 °C·min^−1^. Every 10 °C step, samples were thermostatized for 2 min and four scans were acquired and averaged at that temperature. Once a thermal scan concluded, the sample was cooled down to 5 °C, thermostatized for 5 min, and four scans were registered and averaged. In all cases, scans with buffer and no peptide were registered and subsequently subtracted.

#### 4.2.3. Nuclear Magnetic Resonance (NMR) Spectroscopy

The NMR spectra were acquired at 10 °C on a Bruker Avance 500 MHz spectrometer (Bruker GmbH, Germany), equipped with a triple resonance probe and z-pulse field gradients. Spectra were processed with Bruker TopSpin 2.1 (Bruker GmbH, Karlsruhe, Germany). All NMR experiments with the four peptides were carried out in 100 mM sodium phosphate buffer (not corrected for isotope effects), pH 7.2. Spectra were calibrated with TSP by considering pH-dependent changes of its chemical shifts [61]; probe temperature was calibrated with pure methanol [61].
1D-^1^H-NMR spectra: A total of 48 scans were acquired with 16 K acquisition points for the homonuclear 1D-^1^H-NMR spectra of each isolated peptide at a concentration of 1.2 mM. Water signal was suppressed with the WATERGATE sequence [62]. The spectra were processed after zero-filling and apodization with an exponential window.Translational diffusion NMR (DOSY): The peptide concentrations in DOSY experiments was 100 µM, and 128 scans were acquired, where the gradient strength was varied linearly. Measurements of the translational self-diffusion were performed with the pulsed-gradient spin-echo sequence in the presence of 100% D_2_O. Details on the experimental conditions and fitting of the resulting curves have been described elsewhere [42]. The gradient strength was varied in sixteen linear steps between 2 to 95% of the total power of the gradient coil. Gradient strength was calibrated by using the value of the translational diffusion coefficient, *D*, for the residual proton water signal in a sample containing 100% D_2_O, in a 5-mm tube [63]. The length of the gradient was 2.5 ms; the time between the two pulse gradients in the pulse sequence was 250 ms; and the recovery delay between the bipolar gradients was 100 µs. The methyl groups between 0.8 and 1.0 ppm were used for peak integration for both peptides. A final concentration of 1% of ultra-pure dioxane, which was assumed to have a hydrodynamic radius *R*_h_ of 2.12 Å [63], was added to the solution.2D-^1^H-NMR spectra: Two-dimensional spectra of the four peptides in aqueous solution (100 mM phosphate buffer, pH 7.2) or in the presence of deuterated TFE were acquired in each dimension in phase-sensitive mode by using the time-proportional phase incrementation technique [64] and a spectral width of 5500 Hz; the concentration of the peptide was the same used in the 1D-^1^H-NMR experiments. Standard TOCSY (with a mixing time of 80 ms) [65] and NOESY experiments (with a mixing time of 225 ms) [66] were performed by acquiring a data matrix size of 4096 × 512 points. The DIPSI (decoupling in the presence of scalar interactions) spin-lock sequence [67] was used in the TOCSY experiments, with a relaxation time of 1 s. A number of 96 scans were acquired *per* increment in the first dimension, and the residual water signal was removed by using the WATERGATE sequence [62]. NOESY spectra were collected with 128 scans per increment in the first dimension, again using the WATERGATE sequence [62], with a relaxation time of 1 s. Data were zero-filled, resolution-enhanced with a square sine-bell window function optimized in each spectrum, and baseline-corrected. The ^1^H resonances were assigned by standard sequential assignment processes [43]. The chemical shift values of H_α_ protons in random-coil regions were obtained from tabulated data, corrected by neighboring residue effects [43,45,46].


#### 4.2.4. Statistical Mechanical Calculations

The conformational features of the peptides are predicted by the exact-solution to the Wako-Saitô-Muñoz-Eaton (WSME) model that assumes an ensemble of *2^N^*, where *N* is the peptide length [68,69]. In the current work, the model assumes a large ensemble of 4,194,304 microstates corresponding to the 22-residue peptide. Native interactions were identified with a 6 Å heavy-atom distance cut-off excluding nearest neighbors, while charge–charge interactions were considered, without assuming any distance cut-off. The two thermodynamic parameters fixed across all the peptides are: van der Waals interaction energy per native contact of −80 J·mol^−1^ and an entropic penalty for fixing a residue in the native conformation of −16.5 J·mol^−1^·K^−1^ per residue for all residues, other than proline, glycine, and non-helical residues. All non-helical and glycine residues were provided an additional entropic penalty of −6.06 J·mol^−1^·K^−1^ per residue [70], given their larger degrees of freedom, while proline was assigned zero entropic penalty, given its limited conformational flexibility. One-dimensional free energy profiles and two-dimensional landscapes were generated by accumulating partial partition functions involving a specified number of structured residues.

### 4.3. Characterization of the Effects of T9E and T9A on Full-Length AGT Proteins 

#### 4.3.1. Protein Expression and Purification

Mutations to T9E and T9A were introduced into pCOLDII plasmids containing AGT WT, LM, LM-G170R, or LM-I244T [31] using standard mutagenesis protocols and confirmed by DNA sequencing of the entire cDNA. BL21(DE3) *E. coli* strains were transformed with these plasmids, and protein expression was induced by addition of 0.4 mM IPTG for 4–6 h at 4 °C. AGT proteins were purified from soluble extracts by metal affinity chromatography and subsequent size exclusion chromatography, as described in [31,32]. Apo proteins were produced by treatment with L-Ala and exposure to a mild acidic pH (250 mM Na-phosphate, pH 5.8), as previously described [32]. Proteins were stored in Na-HEPES 20 mM NaCl 200 mM pH 7.4, and their concentration was measured spectrophotometrically using ε_280_ = 47,000 M^−1^·cm^−1^ for the AGT monomer. Unless otherwise indicated, all subsequent experiments were carried out in the same buffer. 

#### 4.3.2. Overall Transaminase Activity

AGT-specific activity was measured at 37 °C, typically using 2.5 (32 nM) to 4 (51 nM, both in subunit) µg·ml^−1^ purified AGT protein. The reaction mixture contained 10 mM glyoxylate, 100 mM L-alanine, and 150 µM PLP in 100 mM K-phosphate pH 8. The reaction time was set to 2 min and the reaction quenched by using trichloroacetic acid (25% *w/v* final concentration). Samples were clarified by centrifugation at 21,000 g for 10 min at 4 °C, and supernatants were stored at −20 °C. Pyruvate formed in these reactions was determined using lactate dehydrogenase and NADH 0.2 mM at 37 °C. The reaction was monitored as the change in A_340 nm_ in TRIS-HCl 1 M pH 8 using quartz cuvettes in an Agilent HP8453 spectrophotometer at 25 °C. These amounts of pyruvate were determined using calibration curves of pyruvate under the same conditions. For each variant, we carried out at least five different replicas, and the results were presented as mean ± s.d. 

#### 4.3.3. Spectroscopic and Light Scattering Experiments

All spectroscopic measurements were carried out in Na-HEPES 20 mM NaCl 200 mM pH 7.4, unless otherwise indicated. UV-visible absorption spectra were collected in an Agilent HP8453 diode-array spectrophotometer at 25 °C using 1 cm path length cuvettes and 20 µM (in subunit) AGT proteins, as purified. To trigger the formation of PMP, a final concentration of 0.2 M L-alanine was added, and the reaction was allowed to proceed for at least 10 min at 25 °C before spectra collection. To determine whether the PMP formed was released upon addition of L-alanine, these reactions were filtered using 30 kDa cut-off concentrators (VIVASPIN^®^6, Sartorius), and the spectra of filtrates was acquired under similar conditions to those used with AGT proteins. CD measurements were performed at 25 °C in a Jasco J-715 spectropolarimeter equipped with a Peltier temperature controller. AGT samples for CD measurements in the near UV-visible range were prepared as described above for absorption measurements. Spectra were recorded at a scan rate of 100 nm·min^−1^ with a 2 nm bandwidth, and five scans were registered and averaged using 5 mm path-length cuvettes, and appropriate blanks without protein were registered and subtracted. For far-UV CD measurements, AGT proteins were prepared at 5 µM (in subunit) in K-phosphate 20 mM pH 7.4, and spectra were recorded at a scan rate of 100 nm·min^−1^ with a 1 nm bandwidth, and six scans were registered and averaged using 1 mm path-length cuvettes, and appropriate blanks without protein were registered and subtracted. Absorption and CD spectra were recorded from two different preparations of each AGT variant (two independent purifications) and averaged. All CD data were normalized for protein concentration using Equation (2). Dynamic light scattering (DLS) was measured using a DynaPro MSX instrument (Wyatt) in a 1.5 mm path-length cuvette and using 10 µM (in subunit) AGT proteins, as purified. Twenty-five spectra were acquired at 25 °C for each DLS analysis, averaged, and used to determine the hydrodynamic radius assuming spherical scattering particles (i.e., the Stokes-Einstein approach). Radii were expressed as mean ± s.d. from 4–6 replicates.

PLP binding to apo proteins was measured under pseudo first-order kinetic conditions. PLP (5–100 μM) was always in a large excess, compared to apo-AGT (typically 0.4 μM in monomer). Protein samples were thermostatized in 1 cm quartz-cuvettes at 25 °C for five minutes, and PLP was added to an appropriate final concentration and mixed manually (the dead time was 10–40 s, as registered and appropriately considered in the calculations). Time-dependent emission fluorescence was acquired in a Cary Eclipse spectrofluorimeter (Varian, Agilent, Madrid, Spain), using an excitation wavelength of 280 nm and an emission wavelength of 340 nm (5 nm slits) at 25 °C. Kinetic traces were fitted to a single exponential function to provide the observed rate constant *k*_obs_. Under pseudo first-order conditions, this rate constant is *ideally* related to the association equilibrium constant *K*_a_, the second-order association rate constant *k*_on_, and the first-order dissociation rate constant *k*_off_, as follows:(4)$$k_{obs}=k_{off}+k_{on}\cdot \left[PLP\right]$$
(5)$$K_{a}=\frac{k_{on}}{k_{off}}$$
where [PLP] is the total PLP concentration. Thus, a linear fit of Equations (4) and (5) to the *k*_obs_ vs. [PLP] data provides the values of the apparent equilibrium and kinetic constants for PLP binding, as well as their corresponding fitting errors. Dissociation constants (*K*_d_) were simply calculated, considering that these are equal to the inverse of *K*_a_ (=1/*K*_d_). 

#### 4.3.4. Thermal Denaturation Experiments

Thermal stability of AGT proteins (2 μM in protein subunit) was evaluated by following the changes in fluorescence emission intensity (exc. 280 nm, em. 360 nm; slits 5 nm) using a Cary Eclipse spectrofluorimeter (Varian; Agilent, Madrid, Spain). Holo-AGT samples contained proteins, as purified with 100 μM PLP, while apo proteins were prepared from apo protein stored solutions. All protein solutions were prepared in Na-HEPES 20 mM NaCl 200 mM pH 7.4 and placed in 1 cm path-length cuvettes. After 5 min of incubation at 20 °C, thermal scans were carried out from this temperature up to 70–95 °C at a scan rate of 2 °C·min^−1^. Upon subtraction of pre- and post-transition linear baselines, the apparent denaturation temperature (*T*_m_) was determined as the temperature at which half of the denaturation signal was achieved. Values reported are mean ± s.d. from 4 replicas for each protein variant. 

#### 4.3.5. Aggregation Kinetics Measurements

Aggregation kinetics were measured in an Agilent 8453 diode-array spectrophotometer at 37 °C using 1 cm path length plastic cuvettes and 2 µM (in protein subunit) AGT proteins. A total of 980 µL of Na-HEPES 20 mM NaCl 200 mM pH 7.4 was placed in the cuvettes and thermostatized for 10 min at 37 °C. Then, aggregation measurements were initiated by the addition of 20 µL of 50 µM of holo or apo proteins and rapid mixing with a glass pipette. The apparent absorbance at 400 nm (A_400_) was then monitored for 90 min. Control cuvettes contained only buffer. From these transients, two different parameters were obtained: the maximal A400 after 90′ and the maximal rate (from the first derivative of A_400_ vs. time). Values reported are mean ± s.d. from 4 replicas for each protein variant. 

## Figures and Tables

**Figure 1 molecules-27-08762-f001:**
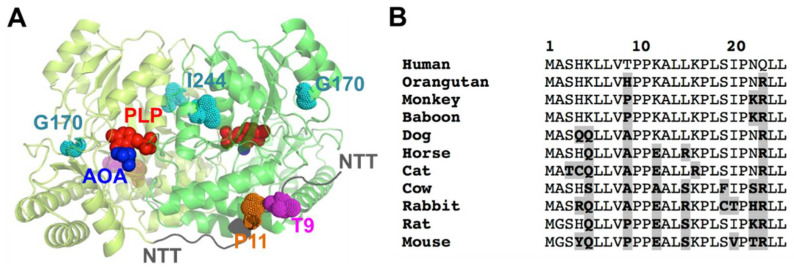
Structure of AGT and the location and conservation of T9 and P11 in the NTT. (**A**) Structural model of the AGT dimer, based on PDB 1H0C [34]. The figure shows the two monomers in the dimer (in two different green tones), the PLP and the inhibitor AOA (amino oxo-acetic acid) as red and blue spheres, respectively; the location of the NTT (in dark grey) with the residues T9 (pink) and P11 (orange) and the PH1 mutated residues in G170R and I244T (in cyan). (**B**) Evolutionary conservation of the 25 N-terminal amino acids of human AGT. Sequences are from mammalian ortologues. Residues in bold and highlighted in grey are those diverging from the human sequence. Note that T9 is only found in the human sequence, whereas P11 is strictly conserved.

**Figure 2 molecules-27-08762-f002:**
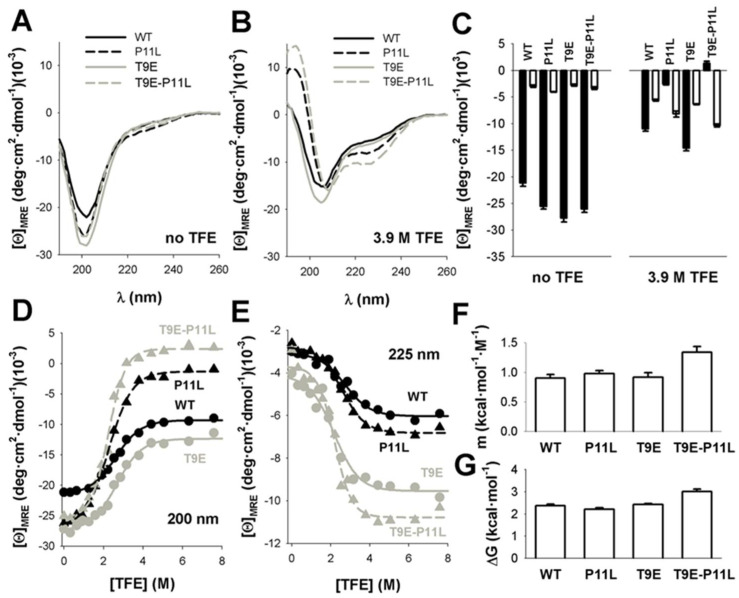
TFE increased ordered secondary structure in NTT −AGT peptides. (**A**,**B**). Far-UV CD spectra of NTT-AGT peptides in aqueous solution (no TFE) or at a high TFE concentration (3.9 M TFE). Spectra are the average of three independent measurements for each peptide and condition. (**C**) Average ± s.d. of mean residue ellipticities at 200 nm (black, indicative of random coil conformations) and at 225 nm (white, indicative of helical conformation). Note that peptides containing P11L and, particularly, T9E-P11L increased the content in ordered structure in the presence of 3.9 M TFE. (**D**,**E**) Quantitative analyses of the TFE concentration dependence of conformational transitions in NTT-AGT peptides at 200 nm (**D**) and 225 nm (**E**). Data are from a single titration for each peptide. Lines correspond to global fits to Equation (2). (**F**,**G**) Equilibrium *m* and Δ*G* values for the conformational transitions induced by TFE, derived from global fits, are shown in panels d, e. The errors in each parameter are the fitting errors to Equation (2).

**Figure 3 molecules-27-08762-f003:**
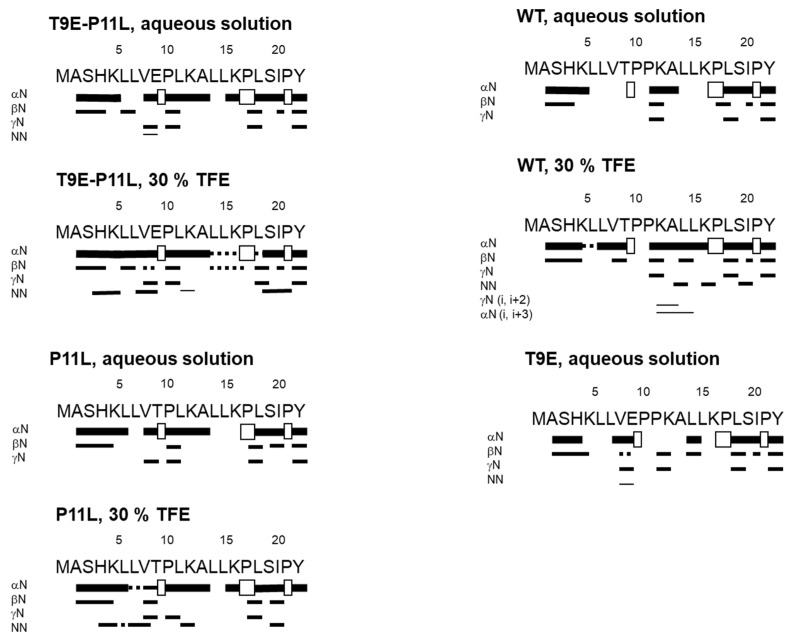
Summary of NMR data for the NTT-AGT peptides in aqueous solution and in 30% TFE (4.2 M). In every panel, NOEs are classified as strong, medium, or weak, according to the height of the bar underneath the sequence; signal intensity was judged by visual inspection from the NOESY experiments with 225 ms of mixing time. The corresponding H_α_ NOEs with the following H_α_ of a proline residue are indicated by an open bar in the row corresponding to the αN (i, i + 1) contacts. The dotted lines indicate NOE contacts that could not be unambiguously assigned, due to signal overlapping.

**Figure 4 molecules-27-08762-f004:**
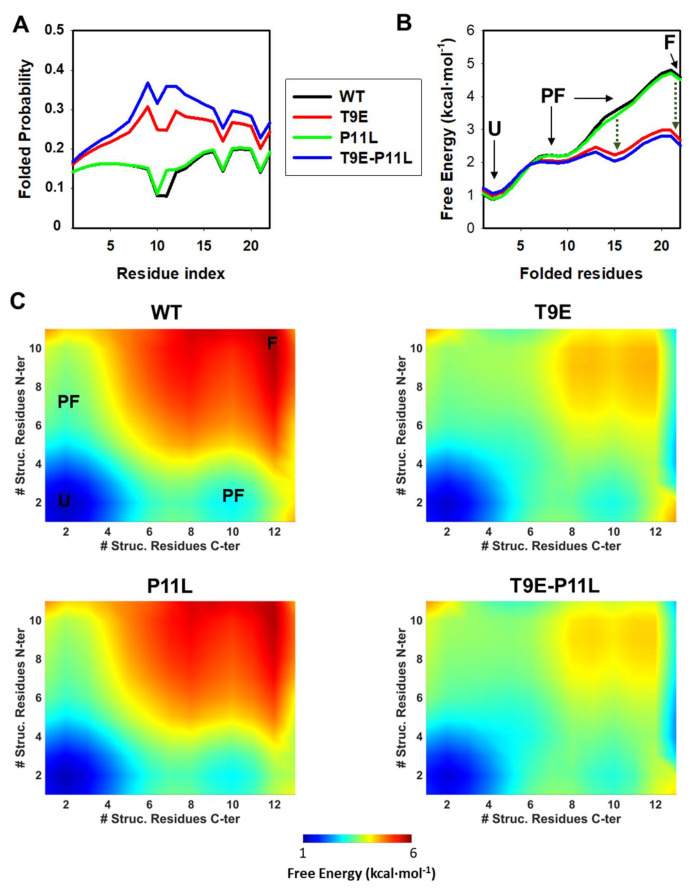
Conformational preferences of NTT-AGT peptides studied by statistical mechanical models. (**A**) Residue level degree of folding (i.e., residue folded probability). (**B**) One-dimensional free energy profiles as a function of the reaction coordinate (i.e., number of folded residues). Four macrostates are observed: one unfolded (U), two partially folded (PF) and a folded (F) state. Stabilization of a PF and the F state by T9E is indicated by vertical, dark green dotted lines. (**C**) Two-dimensional landscapes for the NTT-AGT peptides. The color scale gradually displays the free energy level (from low in blue to high in red). U, PF, and F macrostates are indicated for NTT-AGT-WT. The x and y axis represent the number of structured residues in the N- and C-terminal halves. All calculations were performed at pH 7.0 and 20 °C.

**Figure 5 molecules-27-08762-f005:**
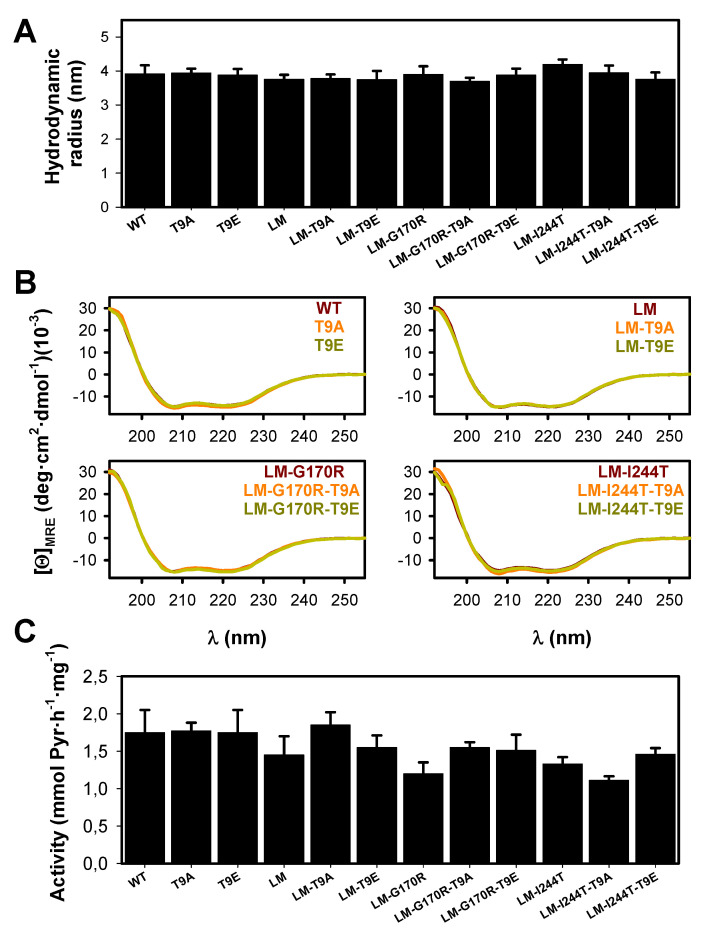
Conformation and activity of AGT variants (WT, LM, LM-G170R and LM-I244T) in the absence or presence of the T9A or T9E mutations. (**A**) Hydrodynamic radii for AGT proteins, as determined by DLS. Data are the mean ± s.d. from three replicates using two different protein preparations. (**B**) Far-UV CD spectra for AGT proteins. Data are the average of two replicas obtained using two different protein preparations. (**C**) Enzyme activity for the overall transamination reaction. Data are the mean ± s.d. from at least five different replicas obtained using two different protein preparations.

**Figure 6 molecules-27-08762-f006:**
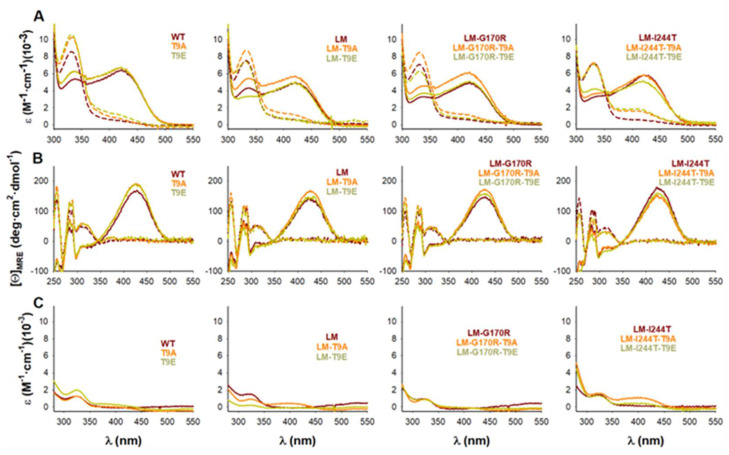
Transamination of bound PLP to PMP in the presence of L-Ala catalyzed by AGT variants (WT, LM, LM-G170R, and LM-I244T) in the absence or presence of the T9A or T9E mutations. (**A**,**B**) Near-UV/visible absorption (**A**) and near-UV CD (**B**) spectra of holo-AGT variants before (solid lines) and after (dashed lines) addition of 100 mM L-Ala. (**C**) Near-UV/visible absorption spectra upon filtration of the reaction mixture (dashed lines in panel **A**) showing the release of PMP from the its tight complex with AGT variants.

**Figure 7 molecules-27-08762-f007:**
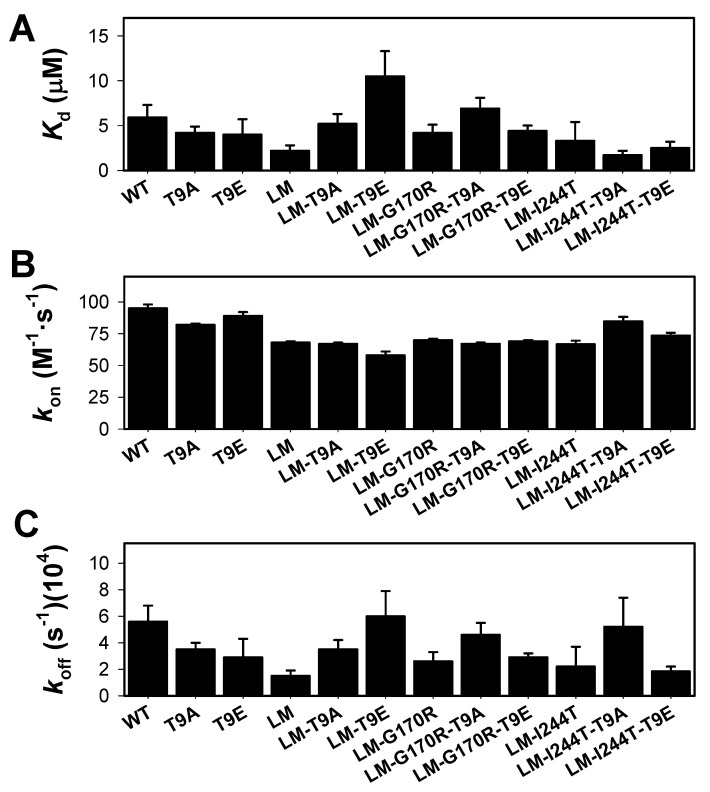
PLP binding to apo-AGT variants (WT, LM, LM-G170R, and LM-I244T) in the absence or presence of the T9A or T9E mutations. (**A**) Apparent dissociation constants for PLP binding; (**B**,**C**) Association (*k*_on_, panel **B**) and dissociation (*k*_off_, panel **C**) rate constants for PLP binding. PLP binding was monitored by quenching of apo-AGT intrinsic fluorescence under pseudo-first order conditions ([PLP] > >[Apo-AGT]). Experiments were performed in HEPES-NaOH 20 mM, NaCl 200 mM, pH 7.4, and 25 °C. Errors are those from fittings or from propagation for *K*_d._

**Figure 8 molecules-27-08762-f008:**
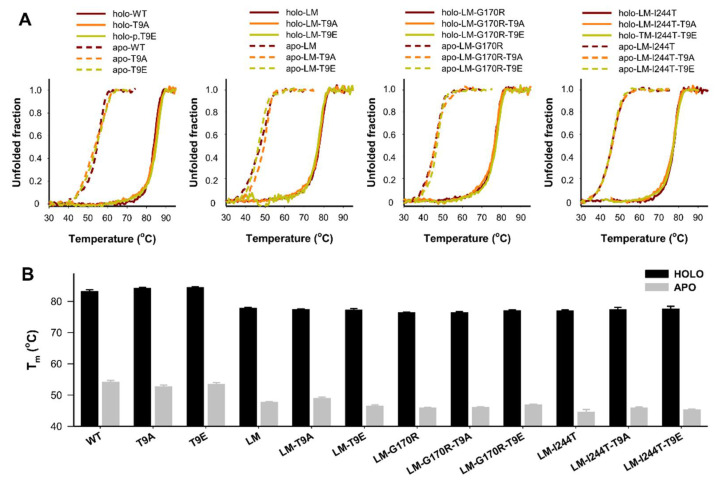
Thermal stability of apo- and holo-AGT variants (WT, LM, LM-G170R, and LM-I244T) in the absence or presence of the T9A or T9E mutations. (**A**) Thermal denaturation profiles monitored by intrinsic emission fluorescence; (**B**) Average (±s.d.) for half-denaturation temperatures (*T*_m_ values) from at least four replicas for each variant. Experiments were performed in HEPES-NaOH 20 mM, NaCl 200 mM, pH 7.4, and 25 °C. Protein concentration was 2 µM (in protein subunits).

**Figure 9 molecules-27-08762-f009:**
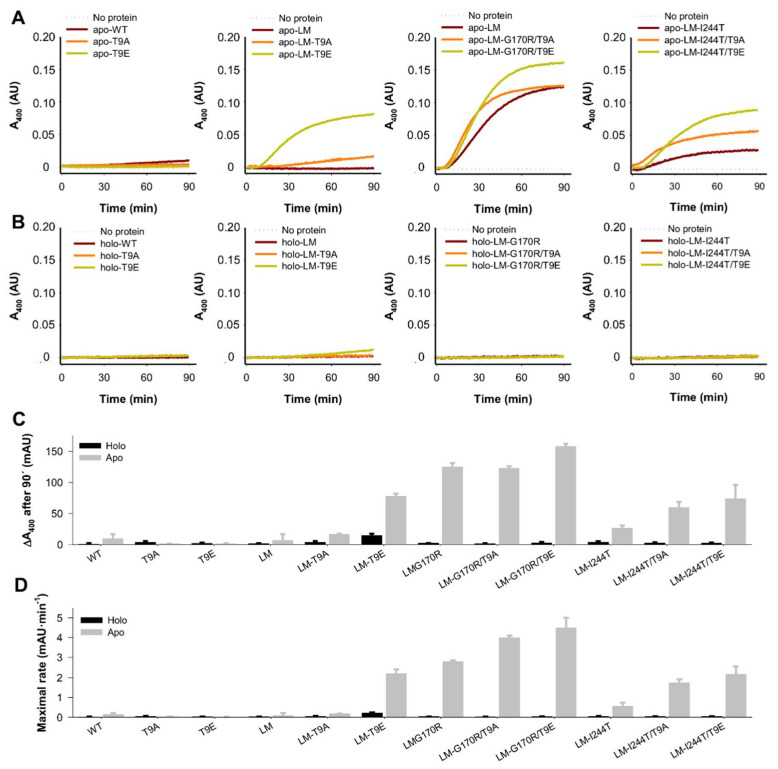
Aggregation kinetics of apo- and holo-AGT variants (WT, LM, LM-G170R, and LM-I244T) in the absence or presence of the T9A or T9E mutations. (**A**,**B**) Time-dependent turbidity (as an apparent A_400_) measurements for apo- (**A**) and holo- (**B**)AGT variants. Data are the average from three replicas. (**C**,**D**) Relevant kinetic parameters, as A_400_ after 90 min (**C**) and the maximum of the first derivative of A_400_ vs. time (**D**). Data in (**C**,**D**) are the mean ± s.d. from four replicas.

**Table 1 molecules-27-08762-t001:** Self-association properties of the peptides in aqueous solution determined by translational diffusion measurements.

NTT-AGT Peptide	*D*, Translational Diffusion Coefficient (cm^2^·s^−1^)	*R_h_,* Hydrodynamic Radius (Å) ^1^
WT	(9.83 ± 0.08) × 10^−7^	12 ± 2
T9E P11L T9E-P11L	(9.69 ± 0.04) × 10^−7^ (9.99 ± 0.01) × 10^−7^ (9.82 ± 0.04) × 10^−7^	12 ± 2 12 ± 2 12 ± 2

^1^ Obtained from the relationship described in [44].

## Data Availability

The datasets generated and/or analyzed during the current study are available from the corresponding author on reasonable request (A.L.P.).

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
