# Peer review of "Phosphorylation of Thr9 Affects the Folding Landscape of the N-Terminal Segment of Human AGT Enhancing Protein Aggregation of Disease-Causing Mutants"

_molecules, 2022, doi:10.3390/molecules27248762_

Round 1
Reviewer 1 Report
Current manuscript ‘Phosphorylation of Thr-9 affects the folding landscape of N-terminal tail of human AGT enhancing protein aggregation of disease-causing mutants’ by Neira et.al., presents an expansion of research into the potential phosphorylation effects of Threonine-9 in AGT function. This article is definitely interesting. Overall the manuscript is well written, literature citation is good and covers many citations. Figures, tables and their legends are self-explanatory and are contiguous with the text. Therefore, I consider this manuscript will be of good interest for the researchers in the community.
Comments that need to addressed:
1. Instead of N-terminal tail, authors should label it as N-terminal segment. Tail more aptly suits for the C-terminal segment of polypeptide chain.
2. Given the presence of at least three proline residues in the N-terminal peptide segments, do authors anticipate it to still form a a-helix. They should explain the reasons for this assumption.
3. N-terminal segment should be highlighted with some color in the figure-1A to make it more apparent.
4. Authors chose to observe helicity by monitoring CD signal at 200 and 225 nm. Traditionally, 222 nm is monitored for helicity. Is there any reason for monitoring CD signal at 225 nm?
5. Typically, alpha helical proteins have less dispersion for the amide protons. Likewise, solvent exposed methyl groups have same chemical shift. Given that they see changes in amide signal dispersion upon addition with TFE, can they introduce supplementary figure representing the increased dispersion of signals in the 2D 1H-1H NOESY spectra. They may also overlay 1D 1H spectra or also to show the variations among different samples will be helpful for the readers.
6. To identify if the observed NOEs represent helical segment or b-turn, authors should convert the NOE peaks into peak intensities or volumes and confirm if they really have b-turn character or helical character.
7. Line-637. Please convert units into micro molar.
8. Line-681. It should be Cary Eclipse
Author Response
1.- Instead of N-terminal tail, authors should label it as N-terminal segment. Tail more aptly suits for the C-terminal segment of polypeptide chain.
We have changed the acronym NTT for N-terminal segment, although we have kept the acronym through the text. We have also changed the title, as indicated in the first paragraph of this letter, when the title was written.
2.- Given the presence of at least three proline residues in the N-terminal peptide segments, do authors anticipate it to still form a a-helix. They should explain the reasons for this assumption.
This assumption is provided in reference [36] and it was already discussed in the Introduction (section 1) and in the Results section (section 2.1.1). We have added a sentence to the latter to pinpoint the argument raised by the reviewer.
3.- N-terminal segment should be highlighted with some color in the figure-1A to make it more apparent.
When preparing Fig.1 we were trying several colors (red and blue among them) to indicate the NTT region, and the figure did not look fine in our opinion (due to the other highlighted residues and especially T9 and P11); that is why we used the grey color to indicate that region, and we prefer to keep that color in the revised version.
4.- Authors chose to observe helicity by monitoring CD signal at 200 and 225 nm. Traditionally, 222 nm is monitored for helicity. Is there any reason for monitoring CD signal at 225 nm?
Reviewer is right, 222 nm is usually monitored to follow a-helix formation as indicated in the reference 39. We followed the changes in ellipticity upon TFE addition at 225 nm, because is the minimum observed at the highest concentration of co-solvent used (as it is clearly observed in Fig. 2 B). Besides, since we are determining ΔG-values, from which we can estimate the helical population (and not absolute values of helicity based on the measurements at a particular wavelength), it is not important at which wavelength we monitored the TFE-transition, as long as the signal-to-noise ratio is good enough. We have written a sentence in Results section (section 2.1.1) to explain the selection of this wavelength.
5.- Typically, alpha helical proteins have less dispersion for the amide protons. Likewise, solvent exposed methyl groups have same chemical shift. Given that they see changes in amide signal dispersion upon addition with TFE, can they introduce supplementary figure representing the increased dispersion of signals in the 2D 1H-1H NOESY spectra. They may also overlay 1D 1H spectra or also to show the variations among different samples will be helpful for the readers.
Reviewer is right, an all α-helical proteins show less dispersion for the amide protons than an α/β, an α + β or an all β-protein. However, the spectra of a-helical proteins show more dispersion than those of disordered proteins, where, in addition the short T2-time (due to their disordered nature) make the signals sharper. Besides, we are working with peptides in aqueous solution, not full proteins, with a rather flexible conformation, and then equilibria between different populations of states (random coil ↔ α-helix ↔ 310-helix ↔ β-turns) must exist resulting in a broadening of the signals. In the presence of the co-solvent, we are shifting those equilibria towards folded conformations, which have a different environment for each amide proton, leading to distinct chemical shifts. Although the dispersion of the amide protons in the presence of TFE, when compared to aqueous solution, can be followed by close inspection of the tables, as requested, we have added as new Fig. S3 of the supplementary Material the regions of the NOESY spectra comprising the NH (F2)-Ha (F1) for the wild type in water and in the presence of the co-solvent. We have also added as new Fig. S4 the amide regions in the presence of the co-solvent for the four peptides, and therefore the reader can compare these 1H-1D-NMR spectra with those of Fig. S2 (top section). The rest of the figure numbering in the SM has been updated through the text; the final paragraph of the manuscript describing the Supplementary Material contents has also been updated.
6.- To identify if the observed NOEs represent helical segment or b-turn, authors should convert the NOE peaks into peak intensities or volumes and confirm if they really have b-turn character or helical character.
We tried to do that, but the overlapping of the amide protons in the NOESY spectra (as it was stated in the old version of the manuscript in the Results section, section 2.1.2.) precluded a reliable estimation of the intensities of the intra-residue aN(i, i) and the sequential aN(i, i+1) NOEs. That is why we preferred to be more conservative about the predominant structure indicating both possibilities in our Discussion of the Results.
7.- Line-637. Please convert units into micro molar.
We have converted it.
8.- Line-681. It should be Cary Eclipse
We apologize for that mispelling of ours. We have corrected it.
Reviewer 2 Report
The work presented in the paper submitted by Neira et al. describes an in-depth analysis of the role played by the phosphorylation of Thr9 (mimicked by the Thr9 to Glu mutation) on the folding, stability and aggregation properties of human AGT, a protein involved in primary hyperoxaluria (PH1). A variety of biophysical and computational methodologies were employed to characterize the N-terminal AGT peptide (containing the T9E mutation) and the full-length protein, showing convincing results that the phosphorylation is crucial to modulate the aggregation propensity of some pathological variants of AGT. Proper control experiments have been carried out.
The study is appropriately designed and the conclusions are justified by the experiments.
I therefore suggest the acceptance of the Manuscript.
Author Response
The work presented in the paper submitted by Neira et al. describes an in-depth analysis of the role played by the phosphorylation of Thr9 (mimicked by the Thr9 to Glu mutation) on the folding, stability and aggregation properties of human AGT, a protein involved in primary hyperoxaluria (PH1). A variety of biophysical and computational methodologies were employed to characterize the N-terminal AGT peptide (containing the T9E mutation) and the full-length protein, showing convincing results that the phosphorylation is crucial to modulate the aggregation propensity of some pathological variants of AGT. Proper control experiments have been carried out.
The study is appropriately designed and the conclusions are justified by the experiments.
I therefore suggest the acceptance of the Manuscript.
We thank the reviewer for his/her kind words.
Reviewer 3 Report
This manuscript looks at the effect of phosphorylation of Thr9 of human AGT using a peptide for the N-terminal tail (NTT-AGT) as well as the full-length protein. A T9E mutation is used as phosphomimetic. The work on the NTT-AGT characterizes the synergy between the T9E mutation and a mutation found in the minor allele (LM), P11L. The work on the full-length protein focus on disease-causing mutations, G170R and I244T. With NTT-AGT, the P11L mutation causes some enhancement of structure in the presence of TFE, with clear synergy between T9E and P11L. For the full-length protein, the structure, activity and stability are minimally affected. The primary effect of the T9E mutation appears to be on the aggregation properties of the protein, where it enhances the aggregation of disease-causing variants of apo-AGT. In general, the effects on aggregation are larger in the presence of T9E than for the negative control T9A. The work appears to be carefully done, is clearly presented and conservatively interpreted. A few comments for the authors follow.
1. The authors have measured CD to sufficiently low wavelength that they could readily evaluate the secondary structure content of the NTT-AGT peptides using available methods for estimating secondary structure from CD spectra (diChroWeb, for example). It would be useful to provide these estimates in supplementary material and discuss them in the text.
2. It is well-known that Tyr at the ends of peptides can interfere with CD spectra in the far UV (Biochemistry 1989, 28, 8609-8613; Biochemistry 1993, 32, 21, 5560-5565). It is common to insert a Gly in front of a C-terminal Tyr to prevent this interference, which the authors have not done with their set of peptides. The authors should provide some discussion of how their CD spectra may be affected by the presence of tyrosine at the C-terminus.
3. The authors describe the T9A mutation (lines 283-284) as a negative control. They should be more explicit why this is a negative control.
4. Line 203, give a citation for the isosbestic point of 203 nm for the helix/coli transition.
Author Response
This manuscript looks at the effect of phosphorylation of Thr9 of human AGT using a peptide for the N-terminal tail (NTT-AGT) as well as the full-length protein. A T9E mutation is used as phosphomimetic. The work on the NTT-AGT characterizes the synergy between the T9E mutation and a mutation found in the minor allele (LM), P11L. The work on the full-length protein focus on disease-causing mutations, G170R and I244T. With NTT-AGT, the P11L mutation causes some enhancement of structure in the presence of TFE, with clear synergy between T9E and P11L. For the full-length protein, the structure, activity and stability are minimally affected. The primary effect of the T9E mutation appears to be on the aggregation properties of the protein, where it enhances the aggregation of disease-causing variants of apo-AGT. In general, the effects on aggregation are larger in the presence of T9E than for the negative control T9A. The work appears to be carefully done, is clearly presented and conservatively interpreted. A few comments for the authors follow.
We thank the reviewer for his/her kind words.
1.- The authors have measured CD to sufficiently low wavelength that they could readily evaluate the secondary structure content of the NTT-AGT peptides using available methods for estimating secondary structure from CD spectra (diChroWeb, for example). It would be useful to provide these estimates in supplementary material and discuss them in the text.
We had estimated the secondary structure by using the DichroWeb page (using all the deconvolution programs except VARSLC, for which we need measurements until 178 nm), and we had not included in the Results section because the findings further pinpointing to the mainly random-coil nature of the four peptides. In the revised version, we have included a short paragraph in the Discussion section (section 3), and we have added three new references.
2.- It is well-known that Tyr at the ends of peptides can interfere with CD spectra in the far UV (Biochemistry 1989, 28, 8609-8613; Biochemistry 1993, 32, 21, 5560-5565). It is common to insert a Gly in front of a C-terminal Tyr to prevent this interference, which the authors have not done with their set of peptides. The authors should provide some discussion of how their CD spectra may be affected by the presence of tyrosine at the C-terminus.
We assumed that the presence of a proline residue preceding the tyrosine could have the same effect as the glycine, as described in the paper by Baldwin and co-workers. Besides, it must be stated that as we are determining ΔG-values, from which we can estimate the helical population - and we are not using absolute values of helicity based on the measurements at a particular wavelength -, as long as the changes at a chosen wavelength can be followed with a good signal-to-noise ratio, the influence of the Y bands at any TFE-concentration for any of the peptides would be the same and such presence does not affect the slope of the titration curve (which provides the free energy of the a-helix↔random-coil transition). We have added a paragraph to the Materials and Methods Section (section 4.2.1) and the reference of Baldwin’s work.
3.- The authors describe the T9A mutation (lines 283-284) as a negative control. They should be more explicit why this is a negative control.
The T9A is a negative control, since it does not introduce the phosphomimic glutamic mutation. We have indicated this in Results section (section 2.2).
- Line 203, give a citation for the isosbestic point of 203 nm for the helix/coli transition.
The reference is that previously cited at the beginning of such paragraph: 40. We have added this reference at the end of this paragraph.